# Innovative Seatbelt-Integrated Metasurface Radar for Enhanced In-Car Healthcare Monitoring

**DOI:** 10.3390/s24237494

**Published:** 2024-11-24

**Authors:** Rifa Atul Izza Asyari, Roy B. V. B. Simorangkir, Daniel Teichmann

**Affiliations:** 1SDU Health of Informatics, The Maersk Mc-Kinney Moller Institute, Faculty of Engineering, University of Southern Denmark, 5230 Odense M, Denmark; 2Department of Engineering, Durham University, Durham DH1 3LE, UK; roy.b.simorangkir@durham.ac.uk

**Keywords:** radar, metasurface, healthcare, seatbelt

## Abstract

This study introduces a novel seatbelt-integrated, non-invasive, beam-focusing metamaterial sensing system characterized by its thinness and flexibility. The system comprises a flexible transmitarray lens and an FMCW radar sensor, enabling the accurate detection and analysis of seatbelt usage and positioning through human tissue. The metasurface design remains effective even when subjected to different bending angles. Our system closely tracks heart rate and respiration, validated against standard reference methods, highlighting its potential for enhancing in-car healthcare monitoring. Experimental results demonstrate the system’s reliability in monitoring physiological signals within dynamic vehicular environments.

## 1. Introduction

As the demand for smarter vehicles increases, manufacturers are incorporating a growing number of sensors to enhance functionality and safety [1]. However, the extensive wiring required for these sensors complicates installation and maintenance, particularly for driver and passenger monitoring systems [2]. Current solutions predominantly rely on wired connections [3] or battery-powered wearable sensors [4,5], both of which present significant drawbacks in terms of complexity, reliability, and maintenance.

Far-field radars have been widely used for monitoring physiological parameters, such as heart and lung movements, by detecting changes in electromagnetic waves reflected from a person’s body [6,7]. However, implementing this method in a vehicular environment presents several challenges [8,9]. Engine vibrations and road noise can lead to inaccurate readings, while the movement of the driver and passengers within the vehicle further complicates the radar’s ability to accurately capture vital sign data [10,11].

To tackle these challenges, recent research has explored various approaches. Zheng et al. [12] developed an ultra-wide-band (UWB) radar for detecting drivers’ heart rates (HR) by analyzing Doppler frequency shifts in UWB signals caused by heartbeat, breathing, and ambient noise. Shyu et al. [13] proposed a method that simultaneously detects breathing and heartbeat information from a slightly swinging human (driver) using novel multi-feature alignment (MFA) two-layer ensemble empirical mode decomposition (EEMD). While these techniques offer precise detection, they face limitations such as sensitivity to ambient noise, constraints in sampling rates, and accuracy issues with errors exceeding 50 ms.

Metasurfaces, known for their ability to manipulate electromagnetic waves with high precision, present a promising alternative [14,15]. These engineered surfaces offer unique capabilities in controlling wave propagation, including beam steering, focusing, and shaping of electromagnetic waves. Such properties make metasurfaces particularly suitable for enhancing radar performance in complex environments. However, the application of beam-focusing metasurfaces in vehicular sensor systems requires further investigation to fully assess their effectiveness in in-vehicle applications.

Previous studies have explored the use of metasurfaces for vehicular sensing applications. For instance, Kang et al. [16] developed a seatbelt-integrated multibeam metasurface lens antenna and multifunctional metasurface tags designed to estimate the respiration rates of multiple targets within a vehicle. However, this research did not address heart rate detection or seat occupancy determination. Moreover, it lacked long-term performance results under various driving conditions. Additional research on in-vehicle monitoring radar [17,18,19,20,21,22,23,23] and seatbelt-integrated sensors [24] has been conducted, but limitations persist. These studies primarily focus on a single vital parameter and face challenges in adjusting beam direction and intensity [12], which hinders comprehensive vital signs monitoring in the dynamic and noisy environment of a moving vehicle. The lack of precision impedes reliable detection of target seat occupancy and physiological parameters, such as cardiac and pulmonary movements, leading to inconsistent monitoring outcomes [25,26,27]. The complexity and variability of conditions inside the vehicle, including passenger movement and fluctuating levels of ambient noise, exacerbate these limitations. The referenced studies focus on the development and application of radar-based sensors for various health and occupancy monitoring applications. As to [28,29,30,31,32,33], while these studies provide valuable insights and advancements in signal processing and algorithm development, they often fall short in addressing the fundamental hardware issues that can impact the overall performance and reliability of the sensors. These challenges include signal stability, noise reduction, interference, power management, environmental robustness, and component durability. Addressing these hardware issues is crucial for ensuring consistent and reliable performance of radar-based sensors in real-world applications.

To overcome these limitations, our research introduces a novel seatbelt-integrated, contactless beam-focusing metamaterial sensing system. This system offers several key advantages:Precise beam positioning: Integrating the metasurface into the seatbelt ensures accurate and stable positioning of the radar beam on the driver’s or passenger’s chest. This beam-focusing technology concentrates radar energy on the target area, minimizing background noise and environmental interference. As a result, the system generates a cleaner signal, enabling more reliable detection of subtle variations in physiological parameters, even in the dynamic conditions of a moving vehicle, where maintaining sensor accuracy is challenging.Enhanced sensitivity and selectivity: Our system leverages the unique properties of metamaterials to enhance the sensitivity and selectivity of the radar. These engineered surfaces enable tight beam focusing and couple confined electromagnetic waves with subtle chest movements, allowing for precise detection of cardiac and pulmonary activity compared to prior research [16,34,35].Multiparameter detection: Unlike previous systems limited to detecting a single physiological parameter, our radar sensor system, integrated with metamaterials, can accurately monitor multiple parameters simultaneously. This is achieved through advanced beam-focusing technology, which improves signal precision and reduces interference. As a result, the system can reliably detect various vital signs, even in challenging environments with noise or passenger movement, such as in vehicles. By focusing the radar beam, the system can clearly differentiate between different physiological signals—for example, separating the smaller movements of heartbeats from the larger movements of breathing. This ability to distinguish between signals enables the system to monitor multiple parameters at once.Seat occupancy detection: The integration of metamaterials into the seatbelt allows the system to precisely detect whether a seat is occupied. This ensures that safety features, such as seatbelt reminders and airbag deployment systems, are activated correctly, enhancing overall passenger safety.Advanced signal processing: To effectively process the captured data and accurately extract respiration and heartbeat signals from the complex sensor output, we have employed variational mode decomposition (VMD) [34,35]. This technique enhances the system’s ability to isolate and analyze individual vital sign components.

Variational mode decomposition (VMD) combined with metamaterial design significantly enhances the detection and monitoring of vital signs in challenging vehicular settings. VMD decomposes a signal into distinct modes, each with its own central frequency, allowing it to isolate vital sign components such as respiration and heartbeat from noise and irrelevant signals. This decomposition is particularly effective in vehicular environments, where non-stationary and non-linear signals are common. VMD’s ability to separate noise, which typically spans a broad frequency range, ensures that vital signs can be accurately extracted without interference. Furthermore, VMD distinguishes between stable vital sign frequencies and motion-induced variations, reducing the impact of motion artifacts.

Metamaterials enhance this process by improving the initial signal capture. Engineered to have unique properties, metamaterials in sensors can focus and amplify specific signals, increasing the signal-to-noise ratio and making it easier for VMD to process the data. These materials act as spatial filters, targeting the region of interest, such as the chest area of a passenger, and ignoring irrelevant areas. This spatial selectivity reduces extraneous data and associated noise. Additionally, our metasurface can be easily adapted to different vehicle interiors.

The combination of VMD and metamaterial design results in superior noise filtering and effective motion artifact mitigation. The metamaterial-enhanced sensors capture cleaner signals, which VMD then refines by isolating the respiration and heartbeat components from residual noise. This dual-layer approach ensures high accuracy and reliability in detecting vital signs, even in dynamic vehicular environments where traditional methods often fail. By leveraging the strengths of both technologies, the system provides robust monitoring capabilities, overcoming challenges that have previously limited the effectiveness of vital sign detection in vehicles.

Our approach integrates a flexible transmitarray metasurface into the seatbelts with a 60 GHz FMCW radar system combined with advanced signal processing, representing a major leap forward in non-contact monitoring of drivers and passengers. With its wireless, battery-less, and chipless design, this solution addresses major challenges faced by current radar systems, such as low sensitivity, unstable performance, and noisy waveforms. By enhancing the accuracy and expanding the scope of health monitoring in smart vehicles, our research makes a substantial contribution to the development and innovation of vehicle sensor systems.

## 2. Transmitarray Lens Design

In this study, we use a transmitarray lens to precisely control the electromagnetic wavefront’s phase, amplitude, and polarization, essential for accurately focusing the radar beam on a specific area on the chest. The lens’ unit cell, shown in Figure 1a, is a 1.6 mm square, with approximately $0.6\lambda_{g}$, where $\lambda_{g}$ is the guided wavelength in the polyimide film substrate ($\varepsilon_{r}=3.5$). At the operating frequency 60 GHz, $\lambda_{g}$ is $\lambda_{g}=\lambda_{o}/\sqrt{\varepsilon_{r}}$ = 2.67 mm. This subwavelength unit cell size is crucial for minimizing grating lobes while maintaining a sufficiently wide phase range for effective wavefront manipulation.

The transmitarray lens was designed and optimized using ANSYS Electronics 2023 R2 software employing periodic boundary conditions and Floquet port excitation. The unit cell consists of two layers: An even layer with a square copper pattern and a circular hole and an odd layer with a square copper pattern, as illustrated in Figure 1b. To account for the impact of human tissue on the lens performance, simulations incorporated a multi-layer human tissue model comprising 10 mm of skin $\left(\varepsilon_{r}=35\right)$, 20 mm of fat $\left(\varepsilon_{r}=9\right)$, 20 mm of muscle $\left(\varepsilon_{r}=52\right)$, and 15 mm of bone $\left(\varepsilon_{r}=14\right)$ [36]. This comprehensive model ensures accurate prediction of the lens behavior in close proximity to the human body.

Figure 2a,b present the transmission magnitude and phase characteristics across the 57–62 GHz band as a function of the unit cell dimension *L*. Our primary design objective was to achieve a wide phase range approaching $350^{\circ }$ while maintaining a transmission magnitude above −3 dB. This wide phase range is crucial for effective beam focusing and steering, allowing for precise wavefront manipulation in our transmitarray design. We systematically adjusted *L* from 0.2 to 1.2 mm to explore the range of achievable transmission characteristics. Notably, the wide phase range and high transmission magnitude were consistently maintained across the entire frequency band, demonstrating the broadband capability of our design. This consistency is critical for stable performance across the operational bandwidth of the radar system and highlights the robustness of our unit cell design to frequency variations.

Based on these results, we investigate the concept of non-uniform transmitarray lens configurations. Our goal is to achieve a uniform phase distribution of $0^{\circ }$ across a finite-size transmitarray lens, which is essential for optimal directivity and beam focus. A key challenge is to implement phase corrections by strategically modifying the unit cells. Figure 3a illustrates a carefully designed beam-focusing transmitarray lens, optimized using MATLAB for precise electromagnetic interactions. This advanced design, fabricated on a flexible polyimide substrate as depicted in Figure 3b, consists of 37×37 elements over an area of 59.2 mm×59.2 mm, specifically engineered for 60 GHz operation. This transmitarray lens, operating in the millimeter-wave spectrum, holds significant promise for medical and radar applications, where precise wave focusing and manipulation are crucial.

The tilt angle of the metal-dielectric composite metasurface (MDCM) [37] is related to the phase delay gradient $p_{i}$ introduced in the phase of the electric field as it propagates through the metasurface, given by
(1)$$p_{i}=\sin\delta_{i}\times k_{0}$$
where $\delta_{i}$ is the angle of tilt of the surface, $k_{0}=\frac{2\pi }{\lambda_{0}}$ is the wave number in free space, and $\lambda_{0}$ is the wavelength in free space at the operating frequency.

For focusing the beam at a 0-degree angle, the wave propagates in the direction of the antenna beam, characterized by the projected wave numbers $k_{x}$ and $k_{y}$ along the unit vectors $k_{x}$ and $k_{y}$ in the direction of the unit vectors $\hat{x}$ and $\hat{y}$, respectively. These wavenumbers are related to the phase delay gradients and the physical rotation angles of the MDCMs as follows:(2)$$\begin{array}{cc} & k_{x}=p_{1}\cos\alpha_{1}+p_{2}\cos\alpha_{2}\\ & k_{y}=p_{1}\sin\alpha_{1}+p_{2}\sin\alpha_{2} \end{array}$$

To achieve a beam focus at $0^{\circ }$, the azimuth (ϕ) and elevation angles (θ) are calculated [32] using the wavenumbers
(3)$$\begin{array}{cc} \theta & =\sin^{-1}\left(\frac{\sqrt{k_{x}^{2}+k_{y}^{2}}}{k_{0}}\right)\\ \phi & =\tan^{-1}\left(\frac{k_{y}}{k_{x}}\right) \end{array}$$

Substituting $k_{x}$ and $k_{y}$ from the above equations into the expressions for θ and ϕ, we get the following:(4)$$\begin{array}{cc} \theta & =\sin^{-1}\left[\frac{1}{k_{0}}\sqrt{p_{1}^{2}+p_{2}^{2}+2p_{1}p_{2}\cos\left(\alpha_{1}-\alpha_{2}\right)}\right]\\ \phi & =\tan^{-1}\left(\frac{p_{1}\sin\alpha_{1}+p_{2}\sin\alpha_{2}}{p_{1}\cos\left(↓+p_{2}\cos\alpha_{2}\right.}\right) \end{array}$$

For $0^{\circ }$ beam focusing, θ must be zero, which simplifies the expression for the tilt angles $\alpha_{1}$ and $\alpha_{2}$ such that
(5)$$\sqrt{p_{1}^{2}+p_{2}^{2}+2p_{1}p_{2}\cos\left(\alpha_{1}-\alpha_{2}\right)}=0$$

This condition implies that the phase delay gradients must cancel each other out, and thus the MDCMs need to be aligned appropriately [38]. Using these relationships, the elevation and azimuth angles of the output beam can be analytically determined, ensuring that the beam focuses at a angle $0^{\circ }$ by properly selecting the rotation angles of the two MDCMs while keeping the base antenna fixed.

Figure 4 (left side) shows the 3D radiation pattern, revealing a highly directional beam with significant gain concentrated along the vertical axis. The narrow and elongated shape of the main lobe suggests a pencil beam, which is advantageous for targeting specific areas with minimal spread. The color gradient ranges from red (0 dB) at the highest gain to blue (−30 dB) at the lowest, the red region at the top indicating the area of maximum gain, essential to achieving high precision and accuracy in target detection [39,40]. Figure 4 (right side) illustrates how the gain is distributed radially from the center. This pattern helps to understand how the energy disperses on a plane perpendicular to the beam’s main axis. The concentric rings with varying colors indicate different gain levels, with the central red region representing the highest gain, surrounded by rings of decreasing gain, ensuring consistent performance in different orientations.

To design a greater performance metasurface beam focusing with normal incident, the following phase profile of hyperbolical expression along one direction should be satisfied, i.e.,
(6)$$\varphi \left(x\right)=\frac{2\pi }{\lambda }\left(\sqrt{x^{2}+F^{2}}-F\right)$$
where λ is the working wavelength, F is the designed focal distance, and φ(x) is the phase distribution in the coordinate related to location *x*.

The provided figures illustrate the focusing performance of a 60 GHz metasurface lens with dimensions of 37×37 elements. The analysis focuses on the normalized power distribution along the z-axis and the full width at half maximum (FWHM) [41] of the focused beam along the x-axis. In Figure 5a, the intensity profile of the power along the z-axis demonstrates the behavior of the focused beam as it propagates. The profile shows an initial decrease in intensity close to the lens, followed by a gradual increase, peaking around 150 mm, corresponding to the focal length. This peak signifies the point of maximum focus, after which the intensity drops off, indicating the beam’s divergence beyond the focal point. This behavior confirms the effective convergence of the beam by the metasurface elements, which manipulate the incoming waves to focus them precisely.

Figure 5b analyzes the beam width through the FWHM, a critical metric that quantifyes the beam width at half of its maximum intensity. The bell-shaped curve in the profile, with maximum intensity in the center (x=0 mm) and symmetrical falloff on both sides, indicates a well-focused beam. The measured FWHM is approximately 20 mm, reflecting the beam’s lateral resolution. A smaller FWHM value is desirable for a tightly focused beam, which is beneficial for applications requiring high spatial resolution. The symmetry of the intensity profile suggests a uniform and well-focused beam, which is crucial for consistent performance across the focal plane. The performance characteristics demonstrated by the metasurface lens at 60 GHz have several significant implications [42]. In high-resolution imaging applications, such as medical imaging and precision radar systems, the sharp focus and small FWHM enable detailed imaging of small objects or fine features [43]. For sensing applications, a tightly focused beam can improve detection sensitivity and accuracy, especially in scenarios requiring pinpoint targeting or coverage in a small area [44]. The enhanced focus at 60 GHz is advantageous for material characterization and surface profiling, where detailed spatial information is critical.

The coordinates of any element on the focused transmitarray can be expressed as $\left(x_{ij},y_{ij}\right)$, and the focal length is recorded as $F_{t}$. The transmission phase difference $\Delta \varphi_{i}$ for the *i* th element is shown in Equation (Equation 7).
(7)$$\Delta \varphi_{i}=2\cdot k\cdot \left(\sqrt{R_{i}^{2}+F_{t}^{2}}-F_{t}\right)$$
(8)$$R_{i}=\sqrt{x_{ij}^{2}+y_{ij}^{2}}$$
where *k* is the propagation constant and Equation (Equation 8) denotes the distance from the center of the focused transmit array to any element [45]. The color gradients are indicative of the intensity of the electric field within the tissue, measured in volts per meter (V/m).

Figure 6 illustrates the electric field intensity distributions for a flexible transmitarray lens subjected to bending angles of $10^{\circ },20^{\circ },30^{\circ }$, and $40^{\circ }$. These simulations account for the propagation of electromagnetic waves through human tissue, which has a significant impact due to its high dielectric constant ($\epsilon_{r}\approx 48$). This property affects the speed of the wave and alters the wavefront, with the speed in the tissue reduced to $c_{\mathrm{tissue}}\;=c/\sqrt{\epsilon_{r}\mu }$ and the number of waves defined as $k_{\mathrm{tissue}}\;=$$\frac{2\pi f}{q_{\mathrm{iseser}}}$. A transmitarray lens, composed of subwavelength elements, manipulates the phase of transmitted waves to focus them effectively. When the lens is bent, the wave paths change, introducing phase variations that affect the focusing ability. The bending angle influences the effective focal length and thus the phase distribution and focusing performance.

At a bending angle of $10^{\circ }$, the electric field shows a nearly ideal concentric pattern with a distinct focal point in the center, indicating strong constructive interference and effective focus. This suggests that minor bending introduces minimal phase errors and maintains adequate performance. At $20^{\circ }$ bending, the concentric wavefronts remain mostly intact, although the intensity at the focal point becomes more diffuse, implying that phase coherence is slightly affected. The lens still manages to maintain its focusing capability, though with a slightly broader field distribution. As the bending increases to $30^{\circ }$, the plots begin to show noticeable irregularities in the wavefronts. The focal point is less pronounced, highlighting that phase errors introduced by bending are significant enough to reduce the sharpness of focusing. This suggests a decrease in constructive interference due to more substantial deformation of the wavefronts. At the highest bending angle of $40^{\circ }$, the field distribution shows the most deviation from an ideal pattern. The focal point is less defined, and the intensity spreads over a broader area, indicating that the phase errors are substantial, causing decreased focusing performance. The distorted wavefronts at this level of bending show the limits of the lens’s phase control, as the cumulative phase path differences disrupt the coherence required for strong focusing.

In general, the analysis of Figure 6 demonstrates the impact of bending on the focusing ability of a flexible transmitarray lens. Although minor bending ($10^{\circ }$ to $20^{\circ }$) maintains the lens’s focusing ability with only slight phase deviations, more substantial bending $\left(30^{\circ }\right.$ to $\left.40^{\circ }\right)$ introduces significant phase errors that diminish constructive interference, leading to a broader and less-defined focal point. This analysis underscores the need for robust phase management and adaptive compensation mechanisms to maintain the performance of flexible metasurfaces when subjected to mechanical deformations.

Figure 7 presents an analysis of the implementation of a flexible metasurface integrated into seatbelt materials, emphasizing how the total thickness of the initial matching layer impacts the reflection coefficient. This analysis identifies the conditions that promote effective transmission and minimize reflection. In Figure 7a, a broad range of permittivity values, from 2 to 2.5, and thicknesses, between approximately 1.13 × 10^−3^ m and 1.15 ×10^−3^ m, are analyzed. The gradient color scale represents the reflection coefficient, where lower reflection values are shown in cooler colors (blue and cyan), and higher values are depicted in warmer colors (yellow, orange, and red). This broader analysis illustrates how different combinations of permittivity and thickness affect wave reflection and transmission.

Figure 7b refines the analysis to a more targeted range, with permittivity values between 2.28 and 2.3 and thicknesses from about 1.137 × 10^−3^ m to 1.139 × 10^−3^ m. This narrower focus reveals an overall lower reflection coefficient, indicated by the predominance of cooler colors (mainly blue to cyan). This suggests that within this range of permittivity and thickness, optimal conditions for wave transmission are achieved, significantly reducing reflections. This insight is critical for practical applications involving seatbelt materials and similar multi-layered structures, as it highlights the importance of precise control over material properties and thicknesses. The findings also underscore that despite variations in materials, such as different types of fabric, padding, or even air gaps between layers, optimal transmission can still be achieved when we implement a flexible metasurface transmitarray.

Figure 8 provides a comprehensive analysis of the performance metrics of a flexible metasurface under varying bending angles. Three key aspects are examined: focal shift, full width at half maximum (FWHM), and signal-to-noise ratio (SNR). Figure 8a illustrates the relationship between the focal shift and bending angle. It is evident that the focal shift remains relatively stable and close to zero across all tested bending angles (0 to 40 degrees). This consistency indicates that the metasurface maintains its focal point effectively even as it undergoes deformation. The robustness in focal shift suggests that the metasurface’s design is resilient to bending, ensuring that the lens maintains its focusing capabilities without significant displacement of the focal spot.

Figure 8b presents the FWHM versus bending angle. Similar to the focal shift, the FWHM remains consistent and does not display significant variation across the bending angles. This result implies that the beam width and focusing performance are not compromised as the metasurface bends. The stability in FWHM suggests that the metasurface can preserve its optical focusing characteristics and maintain a sharp focal point, even under mechanical deformation. Such performance is desirable for practical applications where flexible and conformal metasurfaces need to maintain precise focusing abilities under various conditions. Figure 8c shows the SNR as a function of the bending angle. Unlike the focal shift and FWHM, the SNR exhibits a clear downward trend as the bending angle increases. The steady decrease in SNR indicates that, while the metasurface can maintain focal integrity and beam width, there is a gradual decline in signal quality as bending intensifies. This behavior can be attributed to phase distortions and an increase in noise as the metasurface bends, leading to reduced signal coherence. The decline in SNR highlights the impact of mechanical deformation on the overall quality of the signal, suggesting that while the metasurface is resilient in terms of focusing, its ability to maintain a high signal-to-noise ratio may be affected by larger bending angles.

The results demonstrate that the flexible metasurface maintains stable focal shift and beam width (as indicated by the FWHM) across a range of bending angles, showcasing its robustness in focusing performance. However, the SNR analysis reveals that the quality of the signal degrades with increased bending, signaling potential challenges in maintaining high signal fidelity under significant deformation. This analysis underscores the importance of designing flexible metasurfaces with materials and structures that can mitigate noise and phase distortion effects, especially when applications require high SNR.

## 3. Radar System Integration and Implementation

In this study, we integrated the BGT60TR13C FMCW radar sensor, operating in the frequency range of 58 to 63.5 GHz, with a custom metasurface transmitarray lens. The transmitarray lens was strategically attached to the seatbelt, as shown in Figure 9. To evaluate the performance of this setup, experiments were conducted with five healthy adult subjects, aged 25 to 40 years, who participated voluntarily and anonymously after providing informed consent. Each subject was seated in a standard car seat with the seatbelt fastened, ensuring consistent experimental conditions. The transmitarray lens was affixed to the seatbelt, while the radar sensor was positioned in front of the subject at a distance of 0.6 m, aligned with the radar’s optimal detection range.

Data collection for each subject was conducted over a 5-minute period to assess the radar’s ability to reliably track micro-movements, such as breathing patterns. The experimental protocol consisted of three distinct phases: baseline measurement, controlled interference, and posture variations. During the baseline measurement phase, data were collected with the subject seated and wearing typical indoor clothing to replicate real-world conditions. In the controlled interference phase, additional data were collected with a layer of material (e.g., light clothing or a simulated seatbelt pad) placed between the metasurface and the subject to evaluate the impact of intermediate layers on signal quality. Finally, during the posture variations phase, subjects made slight adjustments in their seating position to test the radar’s ability to maintain accurate detection despite minor changes in position.

The BGT60TR13C sensor provides high-precision measurements due to its frequency-modulated, continuous-wave (FMCW) technology [46], Throughout the experiment, each subject’s data was analyzed to identify any deviations caused by movement or environmental factors. Table 1 outlines the key parameters used in the FMCW radar configuration, including the number of ADC samples (64 samples per chirp), the number of chirps (1), chirp time 32 μs, 1 transmit and 3 receive antennas, total bandwidth of 5 GHz, frame time (75.476 ms), and azimuth antenna field of view $70^{\circ }$. These parameters were chosen to enhance the radar’s real-time monitoring capabilities and directional detection performance [47].

Figure 10 illustrates the signal processing block diagram for the FMCW radar system. The process begins with transmitting the FMCW signal and receiving the reflected signal. The received signal is then mixed with the transmitted signal and passed through a low-pass filter. The signal then undergoes quadrature mixing and analog-to-digital conversion (ADC). After downconversion, beam separation is performed, followed by segmentation of time periods. Fast Fourier transform (FFT) is applied, and the range bins are extracted to form a distance matrix. The distance matrix is then processed using variational mode decomposition (VMD) [48], a method chosen for its ability to adaptively decompose complex, non-stationary signals into a series of intrinsic modes without requiring prior knowledge of signal components. VMD isolates distinct frequency modes within the distance matrix, which correspond to different physiological activities or environmental signals. This isolation of modes enhances signal clarity, allowing for more accurate analysis of specific components within the data. Following VMD, phase extraction and frequency analysis are performed to identify low-frequency components associated with vital signs, such as respiration and heartbeat. These components are separated from noise and interference, improving the accuracy and reliability of vital sign detection. The final results, which include real-time monitoring of physiological signals, are displayed for observation, allowing for continuous and noninvasive monitoring of vital signs.

The lightweight and flexible nature of our transmitarray lens allows it to bend between $5^{\circ }$ and $15^{\circ }$ without affecting antenna performance or vital sign detection accuracy. The transmitted signal from the FMCW radar is defined as [49] follows:(9)$$S_{TX}\left(t\right)=\exp\left[j\left(2\pi f_{c}t+\pi \frac{B}{T_{f}}t^{2}+\varphi \left(t\right)\right)\right]$$
where $f_{c}$ denotes the center frequency, *B* represents the signal bandwidth, $T_{f}$ is the duration of the ramp, and φ(t) indicates the initial phase of the signal [50]. Given the seatbelt position, the displacement between the antenna and the human body due to respiration and heartbeat is modeled as
(10)$$R\left(\tau \right)=d_{0}+A\sin\left(2\pi f_{r}\tau \right)+B\sin\left(2\pi f_{h}\tau \right)$$
where $d_{0}$ is the initial distance between the antenna and the body; *A* and *B* are the peak amplitudes of respiratory and heartbeat-induced displacements, respectively; and $f_{r}$ and $f_{h}$ are the frequencies of respiration and heart rate [51].

The signal received back at the radar, considering a time delay $t_{d}$, is
(11)$$\begin{array}{cc} S_{RX}\left(t\right) & =\exp\left[j\left(2\pi f_{c}\left(t-t_{d}\right)+\pi \frac{B}{T_{f}}{\left(t-t_{d}\right)}^{2}+\varphi \left(t-t_{d}\right)\right)\right]\\ t_{d} & =\frac{2R(\tau )}{c} \end{array}$$

The resulting signal x(t), formed by the convolution of $S_{TX}\left(t\right)$ and $S_{RX}\left(t\right)$, simplifies to
(12)$$\begin{array}{cc} x(t) & =S_{TX}\left(t\right)*S_{RX}\left(t\right)\approx \exp\left[j\left(4\pi \frac{BR(\tau )}{cT_{f}}t+\frac{4\pi f_{c}R\left(\tau \right)}{c}\right)\right]\\ & =\exp\left[j\left(f_{b}t+\varphi_{b}\right)\right]\\ f_{b} & =\frac{2BR(\tau )}{cT_{f}}\\ \varphi_{b} & =\frac{4\pi f_{c}R\left(\tau \right)}{c}=\frac{4\pi R(\tau )}{\lambda } \end{array}$$
where *c* is the speed of light, $\varphi_{b}$ indicates the phase of the intermediate frequency (IF) signal [52], and $f_{b}$ is the frequency of the IF signal. The displacement data R(τ), influenced by the movements of the body, are embedded in both $f_{b}$ and $\varphi_{b}$. As the movement of the body caused by heartbeat and respiration is subtle, it cannot be effectively captured by $f_{b}$ due to the resolution limitations of the radar. Consequently, it is derived from $\varphi_{b}$ with greater precision for shorter wavelengths. After quadrature mixing, the intermediate frequency (IF) [50] signal can be approximately expressed as
(13)$$\begin{array}{cc} S_{IF}\left(t\right) & =A_{IF}\exp\left(j\left(2\pi \left[\frac{B}{T_{c}}t_{d}\right]t+2\pi f_{c}t_{d}+\pi \frac{B}{T_{c}}t_{d}^{2}+\Delta \phi \left(t\right)\right)\right)\\ & =A_{IF}\exp\left(j\left(2\pi f_{b}t+\phi_{b}\left(t\right)+\Delta \phi \left(t\right)\right)\right) \end{array}$$
where $A_{IF}$ is the amplitude of the IF signal, and $f_{b}$ is the bandwidth of the IF signal that can be expressed as
(14)$$f_{b}=\frac{2BR(t)}{cT_{c}}$$
and the phase is
(15)$$\phi_{b}\left(t\right)=2\pi f_{c}t_{d}+\pi \frac{B}{T_{c}}t_{d}^{2}\approx 2\pi f_{c}t_{d}$$

The residual phase noise that is Δϕ(t)=ϕ(t)−ϕ(t−2R/c)) can be neglected, and because the time $t_{d}$ is very small, the term $\pi \frac{B}{T_{c}}t_{d}^{2}$ can also be neglected. After sampling, the final signal can be expressed as
(16)$$y\left[n,m\right]=A_{IF}\exp\left(j\left(2\pi f_{b}nT_{f}+\frac{4\pi }{\lambda }\left(nT_{f}+mT_{s}\right)\right)\right.$$
where *n* is the sampling points, $T_{f}$ is the ADC sampling rate, *m* is the sampling points of the slow time, $T_{s}$ is the sampling frequency of the slow time dimension, which is equal to the reciprocal of the frame interval. The frame interval is the duration of time between consecutive radar frames. This setup allows the radar system to capture detailed temporal variations in the signal, which is essential to accurately extract physiological movements such as respiration and heartbeat [53].

To effectively process the captured data and accurately extract the respiration and heartbeat signals from the complex sensor output, we employ VMD. This method allows for the isolation and extraction of subtle physiological signals amidst substantial noise from the vehicle environment. The VMD algorithm operates by iteratively searching for modes that minimize the total bandwidth. This is achieved through the following steps: 1. Initialization: Initialize the modes and their center frequencies. 2. Decomposition: Iteratively update the modes by solving the constrained variational problem to minimize the sum of the bandwidths of the modes. 3. Mode extraction: Extract the modes which represent different intrinsic components of the signal. Mathematically [34,35], the VMD process is described by
(17)$$\min_{u_{k},\omega_{k}}\sum_{k}{\left|\partial_{t}\left[\left(\delta \left(t\right)+j\frac{1}{\pi t}\right)*u_{k}\left(t\right)\right]e^{-j\omega_{k}t}\right|}_{2}^{2}$$
subject to
(18)$$\sum_{k}u_{k}=f$$
where $u_{k}$ are the modes, $\omega_{k}$ are the center frequencies, and *f* is the input signal. By decomposing the radar signal using VMD, we can effectively isolate the components corresponding to respiration and heartbeat. This enables accurate detection of these physiological signals, which previous works could not achieve due to limitations in signal processing and noise reduction techniques.

## 4. Results

Experiments were conducted to validate the practicality and effectiveness of the proposed system. The radar sensor was strategically oriented toward the driver wearing a seatbelt, as shown in Figure 7, to ensure optimal detection of vital signs. The implemented algorithm was applied to process the radar signals, allowing for accurate monitoring of physiological parameters. The radar acquisition range was configured to cover a span of 0.5 to 0.8 m, a distance that is ideal for capturing the necessary data from the driver. This setup was chosen to ensure that the radar could effectively detect subtle physiological movements, such as respiration and heartbeat, without interference from surrounding objects or environmental noise.

To ensure synchronization between radar detection and the standard reference method, specific steps were taken to align the data acquisition and processing timelines of both systems. Time synchronization was established using a unified time source, allowing precise matching of data timestamps from both systems. Simultaneous data acquisition was achieved by triggering both the radar and the reference systems to collect data at the exact moment, ensuring a direct comparison of measurements. Furthermore, a calibration protocol was conducted before each experiment to align radar outputs with standard references, such as the ECG and respiration belt from BIOPAC, ensuring comparable readings of physiological signals. The radar data processing algorithm was further refined to synchronize with the time scale of the reference method, effectively eliminating any potential delay discrepancies. The validation trials subsequently confirmed this synchronization, enabling an accurate overlay of radar and reference data.

The results of these experiments confirmed the ability of the system to operate reliably within the specified range and demonstrated its potential for practical application in real-world driving conditions. Through careful synchronization measures, the system performance was validated against the standard reference method, ensuring accurate and reliable data comparisons.

### 4.1. Seat Occupancy

Radar sensors are crucial for detecting occupants or objects in vehicle seats, ensuring safety and functionality. These sensors identify subtle changes in distance and movement, distinguishing between a person and items. To improve radar accuracy, we are developing a beam-focusing metasurface attached to the seatbelt, which enhances signal reflection. This design boosts the radar’s ability to detect small movements and better differentiate between objects and human occupants, improving overall system reliability and reducing false detections.

When no occupant is present, the radar sensor measures the distance to the seat surface or any objects placed on it, such as a bag. If the object remains still, the sensor detects a relatively constant distance, with only minor fluctuations due to environmental factors such as noise and vibrations within the vehicle, rather than any significant object movement. To determine if a person is present, the standard deviation of the envelope amplitude component is calculated as follows:(19)$$\sigma^{2}\left(t\right)=\int_{0}^{t}{\left(\frac{dy}{dt}\right)}^{2}dt$$

This equation can be implemented digitally using the sampled peak amplitude values, y[n], measured by the radar during each sweep. This type of moving average filter functions similarly to an infinite impulse response (IIR) filter [54]. The formula for implementing Equation (Equation 19) with an exponentially weighted moving average (EWMA) filter is
(20)$$\sigma^{2}\left[n\right]=\alpha {\left(\frac{y[n]-y[n-1]}{T_{s}}\right)}^{2}+\left(1-\alpha \right)\sigma^{2}\left[n-1\right]$$

In this case, the derivative is estimated using the finite difference between two consecutive samples. The smoothing coefficient, α, can be derived from the cutoff frequency [55] using the equation
(21)$$\alpha =\sqrt{\cos^{2}\left(\Omega_{3dB}\right)-4\cos\left(\Omega_{3dB}\right)+3}+\cos\left(\Omega_{3dB}\right)-1$$

Here, $\Omega_{3dB}$ represents the normalized angular frequency, which is given by
(22)$$\Omega_{3dB}=f_{c}\frac{T_{s}}{2\pi }$$

The seat is classified as occupied by a person if the value of σ exceeds a predetermined threshold, $\sigma_{th}$. In the case of a stationary object, the difference between consecutive samples should be zero, with only noise contributing to the signal [56]. Consequently, σ serves as an indicator of noise deviation. The threshold is initially determined on the basis of the average value of σ when the seat is occupied. If the threshold is set too low, it may lead to a higher likelihood of false alarms, while setting it too high could result in detection errors or delayed sensor responses.

Figure 11 illustrates the process of seat occupancy detection using a flexible transmitarray lens integrated with a 60 GHz frequency−modulated continuous-wave radar system (FMCW). The figure contains two key plots. The top plot shows the signal amplitude over a 120 s time interval, capturing both noise and the breathing signal, while the bottom plot displays the seat occupancy indicator, which reflects whether the seat is empty or occupied over time. In the first 20 s, the seat remains empty, as indicated by the low amplitude fluctuating around 0.2 units, suggesting the absence of any significant object. Minor fluctuations are present due to environmental noise, but the system accurately detects the seat as unoccupied. Between 20 and 40 s, a bag is placed on the seat, causing a slight increase in the signal amplitude. However, this increase remains below 0.5 units, indicating that the radar system is capable of distinguishing between an inanimate object, such as a bag, and a person. This reflects the system’s robustness in avoiding false positives for seat occupancy detection.

Seat occupancy detection using 60 GHz FMCW radar, showing range, amplitude (with noise and breathing signal), amplitude deviation, and seat occupancy indicator over time. The green dashed line represents the threshold for detecting seat occupancy, distinguishing between empty, bag placement, and true seat occupancy states. From 40 to 80 s, a person occupies the seat, causing the amplitude to rise sharply to around 0.5 units. During this period, the system detects fine variations in the signal, corresponding to the person’s breathing and slight movements. This demonstrates the radar’s capability to capture micro-movements, which is essential for accurately identifying both presence and physiological signals, like breathing. After 80 s, when the person vacates the seat, the amplitude returns to the lower level, once again signaling an empty seat. The seat occupancy indicator in the second plot also mirrors these changes, clearly switching between empty, bag placement, and occupied states throughout the time interval. This shows the radar system’s ability to reliably differentiate between unoccupied, object placement, and true occupancy by a person, highlighting its sensitivity and accuracy in detecting seat occupancy and physiological movements.

### 4.2. Vital Sign Detection

In this section, we present the findings from our experimental evaluation, which demonstrates the performance and effectiveness of the contactless, beam-focusing, seatbelt-integrated flexible transmitarray metasurface sensing system. The results focus on its ability to accurately detect and monitor vital signs, showcasing its reliability in real-world conditions. Detailed analysis includes comparisons with reference sensor methods, highlighting improvements in signal clarity, response time, and overall data accuracy. These findings underscore the potential of this innovative system for enhanced in-vehicle health monitoring and safety applications.

Figure 12 illustrates the experimental comparison of vital sign detection using a radar system with and without the integration of a flexible transmitarray lens. This experiment, conducted under controlled conditions outside of a vehicle setting, emphasizes the impact of the flexible lens on enhancing the detection of physiological signals. The results clearly demonstrate that the use of the flexible lens significantly improves the accuracy and sensitivity of vital sign monitoring. When the flexible lens is applied, the detected signal amplitudes for both respiration and heart rate are significantly higher and more distinct, underscoring the improved detection capabilities. For instance, the respiration signal exhibits a pronounced peak at 0.49 Hz, and the heart rate signal shows a clearer and more defined peak at approximately 1.04 Hz. These results indicate a stronger and more reliable signal capture when the lens is used, facilitating better identification and analysis of vital signs. Conversely, in the absence of the flexible lens, the amplitudes of the signals for both respiration and heart rate are substantially lower. This reduction reflects a decrease in detection sensitivity and signal clarity, which may hinder the accurate monitoring of vital signs. The loss of signal quality without the lens highlights the potential limitations of using a radar system alone for precise physiological measurements.

Figure 13 presents a comparative analysis of radar-detected vital sign signals against standard reference methods (ECG and Biopac MP160 respiration belt), measured from the same subjects in a nonvehicular experimental setting. This figure is divided into two sections to illustrate the effectiveness of the radar system in detecting both respiratory and heart rate signals. The upper graph displays the respiration signals, with the radar-detected respiration signal (radar resp) shown in blue and the reference respiration belt signal (resp belt) depicted in black. The visual alignment between the radar-detected signal and the respiration belt measurements indicates a strong correlation and high precision in capturing respiratory patterns. This close match suggests that the radar system can reliably monitor respiration rates and patterns comparable to those obtained from the standard respiration belt. The lower graph illustrates the heart rate signals, with the radar detected heart rate (radar heart) in blue and the corresponding ECG measurements in black. Similarly to the respiration signals, the heart rate detected by radar shows a clear and consistent alignment with the ECG signal. This indicates that the radar system is capable of accurately detecting heart rate while maintaining fidelity to the gold standard provided by the ECG measurements.

Figure 14 illustrates the comparison between radar-based measurements and standard monitoring techniques for heart rate and respiration when a flexible transmitarray lens is not used. This experiment, conducted under controlled conditions outside of a vehicle setting. The figure highlights key differences in accuracy and signal stability. In the respiration data, the extracted radar signal follows a waveform similar to the respiration belt, showing that the radar system can reasonably track respiratory activity. However, minor discrepancies in amplitude and slight phase shifts between the two signals indicate that without a flexible lens, the precision of the radar system is somewhat compromised. These variances suggest that, while the radar can detect respiration, its accuracy and reliability are not fully optimal without the lens.

The lower part of the figure presents radar-derived heart rate data compared to the ECG as the gold standard for heart rate monitoring. Unlike relatively accurate respiration tracking, the extracted radar signal shows significant difficulties in capturing heart rate. The radar-based heart rate signal does not replicate the sharp peaks and consistent amplitude seen in the ECG. This lack of precision highlights the system’s limitations in detecting heart rate without the support of a flexible transmitarray lens. The radar signal demonstrates less stability, characterized by significant variations and a more diffuse pattern, indicating difficulties in maintaining consistent detection and alignment with the ECG. The figure demonstrates that without a flexible transmitarray lens, the radar system can still track respiratory activity with moderate accuracy, albeit with some limitations in amplitude and phase stability. However, its ability to accurately detect heart rate is markedly diminished, as evidenced by the poor correlation with the ECG signal. This underscores the importance of a flexible lens for enhancing the sensitivity and stability of radar-based remote monitoring systems, especially for precise heart rate detection.

Figure 15 presents the analysis of the amplitude and frequency of the respiration and heart rate signals under three conditions: car standby, driving, and bumpy condition. The car standby condition with a flexible transmitarray lens demonstrates superior detection capabilities. The time domain signal exhibits clear periodic waveforms, indicating reliable heart rate and respiration monitoring. The frequency-domain analysis supports this observation, showing distinct peaks at 0.45 Hz for respiration and 1.12 Hz for heart rate, aligning closely with the reference ECG measurement at 1.18 Hz. This clear alignment underscores the contribution of the flexible lens to improved system sensitivity, allowing accurate separation of heart rate and respiration signals. In contrast, performance without a flexible transmitarray lens under the same standby conditions shows notable degradation. The time-domain signal appears noisier with less defined patterns, suggesting reduced sensitivity. The frequency domain analysis reveals a weaker heart rate peak at approximately 1.11 Hz, which struggles to stand out against background noise, while the respiration peak at 0.41 Hz remains identifiable but less pronounced. These observations indicate that without the flexible lens, the system’s ability to reliably detect the heart rate is compromised due to insufficient signal clarity and increased noise interference.

Under driving conditions, the system equipped with a flexible transmitarray lens maintains steady detection performance. The time domain signal presents discernible waveforms despite motion, while the frequency domain analysis shows a respiration peak at 0.53 Hz and a heart rate peak at 1.20 Hz, consistent with an ECG reference of 1.21 Hz. This indicates that the flexible lens provides a stabilizing effect that preserves sensitivity even under moderately dynamic conditions, allowing reliable differentiation between heart rate and respiration. However, without the flexible transmitarray lens, the driving condition significantly affects the detection accuracy. The time domain signal becomes more erratic with waveforms that are harder to interpret. The frequency domain analysis shows weaker peaks, with the heart rate appearing around 1.17 Hz but nearly blending into noise, and a respiration peak at 0.51 Hz that is less distinguishable. These results suggest that motion exacerbates the system’s limitations when a flexible lens is not used, leading to challenges in accurately isolating the heart rate signal.

Bumpy conditions pose the most rigorous test for the system. When using a flexible transmitarray lens, the time-domain signal displays more movement-induced disturbances, yet the periodic patterns for respiration and heart rate remain somewhat recognizable. The frequency domain analysis shows peaks at 0.52 Hz for respiration and 1.18 Hz for heart rate, compared to an ECG reference at 1.19 Hz, although with a wider spectral width due to motion artifacts. This shows that the flexible lens effectively enhances the resilience of the system against severe motion, ensuring the detection of vital signs even in challenging conditions. In contrast, without the flexible transmitarray lens, the system’s performance deteriorates considerably in the bumpy condition. The time domain signal becomes highly erratic, with patterns that are difficult to interpret, resulting in unreliable detection. The frequency domain analysis does not show clear heart rate peaks as they are nearly indistinguishable from noise, highlighting the limitations in heart rate detection under high-motion conditions without the stabilizing effect of a flexible lens. Respiration detection also suffers, with peaks at approximately 0.57 Hz that are less defined.

Table 2 provides a detailed understanding of how different driving conditions such as standby, driving, and bumpy conditions affect physiological measurements, specifically heart rate and respiration detection, both with and without the use of a flexible transmitarray lens. The RMS acceleration (RMS Acc.) values indicate the level of environmental vibration experienced during each condition. The data show that using the lens results in lower RMS Acc. values across all conditions: $0.030\;m/s^{2}$ during standby, $0.200\;m/s^{2}$ while driving, and $0.310\;m/s^{2}$ in bumpy conditions. In contrast, without the lens, the RMS Acc. values are higher: $0.037\;m/s^{2}$ in standby, $0.325\;m/s^{2}$ while driving, and $0.470\;m/s^{2}$ in bumpy conditions. These results indicate that the lens helps reduce environmental vibrations, leading to more stable measurements.

The heart rate coefficient of variation (CV HR) also shows improved consistency with the use of the lens. The CV HR values with the lens start at 0.011 in standby, rise to 0.021 during driving, and peak at 0.045 in bumpy conditions. Without the lens, the values are higher: 0.021 in standby, 0.031 during driving, and 0.075 in bumpy conditions. This indicates that the lens helps mitigate motion-induced variability, making heart rate measurements more reliable. The increasing CV HR values across conditions underscore the impact of motion on heart rate detection, particularly in high-intensity conditions like bumpy environments, where the absence of the lens leads to greater variability.

Respiration detection, represented by the coefficient of variation for respiration (CV Resp.), also benefits from the use of the lens, though it remains more stable overall compared to heart rate. The CV Resp. values with the lens are 12.20 in standby, 13.00 during driving, and 14.30 in bumpy conditions, while without the lens, these values are higher: 14.20 in standby, 14.26 during driving, and 15.10 in bumpy conditions. This suggests that while respiration is less affected by motion than heart rate, the use of the lens still contributes to improved consistency in respiration measurements.

Overall, the data indicate that the flexible transmitarray lens enhances the stability and reliability of physiological measurements across all conditions. The lens proves especially beneficial for heart rate detection by reducing variability in high-motion environments. While respiration detection is inherently more resilient to motion artifacts, the lens further improves its measurement consistency. These findings reinforce the value of using the lens for more accurate and stable physiological monitoring in variable driving conditions.

Figure 16 presents a box plot analysis of inter-beat interval (IBI) errors across three different conditions such as standby, driving, and bumpy using of the flexible transmitarray lens. The IBI error is measured in milliseconds (ms) and reflects the deviation of radar−based IBI measurements from the reference standard. The red line within each box plot represents the median value, while the ‘+’ symbols indicate outliers in the data. In the standby condition, the IBI error is relatively low and consistent, with most values clustering around the median and few outliers. During driving, the IBI error distribution is similar to the standby condition but exhibits slightly more variability, indicating minor increases in measurement error due to movement. The bumpy condition shows a significant increase in the variability of the IBI error, with a wider range and higher median error, indicating that rough driving conditions substantially affect the accuracy of radar-based IBI measurements. This analysis highlights the challenges of maintaining precise vital sign monitoring in dynamic and unstable environments. Although this suggests potential limitations in the reliability of the proposed approach for certain applications, it also underscores the need for further development. Defining the severity of “bumpy”conditions and proposing remedies such as enhanced signal processing, multisensor fusion, adaptive calibration, and environmental compensation are crucial steps toward improving robustness. Additionally, understanding the likelihood of encountering such conditions will help in assessing and mitigating these challenges effectively.

The Bland–Altman plots presented in Figure 17 illustrate the agreement between the inter-beat interval (IBI) measurements obtained from the ECG (electrocardiogram) and radar under three different conditions such as car standby (a), driving (b), and bumpy condition (c) using a flexible transmitarray lens. In each graph, the difference between the BBI ECG and BBI radar measurements is plotted against their average, allowing for a visual assessment of the agreement between the two methods. In the car standby condition Figure 17a, the points are relatively tightly clustered around the mean difference line (solid red line), with the majority of differences falling within the limits of agreement (dashed black lines at ±1.96 standard deviations). This suggests a good agreement between the two measurement methods in a stable environment. The estimated radar heart rate for car standby is 72.0661 bpm, compared to the estimated ECG heart rate of 72.6271 bpm, highlighting the minimal discrepancies between the two methods.

Under the driving condition in Figure 17b, there is a slight increase in the spread of the differences between the BBI measurements from ECG and radar. Although the mean difference remains close to zero, the broader range of differences suggests that the agreement between the two methods is slightly less consistent compared to the car standby condition. This increased variability can be attributed to the dynamic nature of driving, which introduces additional noise and movement artifacts. The estimated radar heart rate during driving is 78.7428 bpm, compared to the ECG heart rate of 78.1955 bpm, indicating a small but noticeable variability. Figure 17c shows the highest dispersion of points, with a noticeable trend to increase the difference values as the average BBI increases. This trend, along with the wider spread of points, indicates a reduced agreement between the measurement methods under more challenging conditions with significant motion. The limits of agreement are wider, reflecting the increased variability and potential measurement errors introduced by the bumpy environment. The estimated radar heart rate in bumpy conditions is 84.2739 bpm, while the ECG heart rate is 84.9816 bpm, demonstrating the greatest variability among the conditions tested.

The Bland–Altman plots in Figure 18 illustrate the agreement of the breath–breath interval obtained from the respiration belt and radar under three different conditions such as standby, driving, and bumpy—using a flexible transmitarray lens. In Figure 18a, the plot reveals that the differences between the belt and radar measurements are minimal and centered to be around zero, indicating a negligible systematic bias. The estimated respiratory rates are nearly identical, with the radar measuring 14.9948 breaths per minute and the belt measuring 15.0026 breaths per minute. The majority of the data points fall within the limits of agreement, suggesting a strong correlation between the two methods in a stationary state. For the driving condition depicted in Figure 18b, the Bland–Altman plot continues to show a small mean difference close to zero, with data points closely clustered around the mean difference line. This clustering indicates high consistency and reliability between the belt and radar measurements during driving. The estimated respiratory rates are again very similar, with the radar at 18.001 breaths per minute and the belt at 18.0026 breaths per minute, further underscoring the strong agreement between the two measurement techniques. Under the bumpy condition, as shown in Figure 18c, there is a slight increase in the mean difference, though it remains close to zero, indicating minimal bias even under more dynamic conditions. The estimated respiratory rates are 20.0068 breaths per minute for the radar and 19.9966 breaths per minute for the belt. Although the data points are more spread out compared to the other conditions, they still fall mostly within the limits of agreement. This slight increase in variability can be attributed to increased movement and instability in bumpy conditions, but does not significantly detract from the overall agreement between the two methods.

Figure 19a shows that the error percentages, representing the deviation between the ECG and radar methods, are consistently low in all subjects and driving conditions, generally below 1.5%. This low level of error further supports the accuracy of radar compared to the traditional ECG method. Although there is some variation in the percentages of errors between subjects, with certain individuals showing slightly higher errors under specific conditions, these differences are minimal and do not detract from the overall reliability of the radar method. Additionally, the error percentages remain stable across the various driving conditions, suggesting that the accuracy of radar is not significantly impacted by different types of driving activities, even in more challenging environments like bumpy roads.

The percentages of error in Figure 19b, representing the discrepancies between the belt and radar measurements, are shown for each subject under the three driving conditions. These errors are generally low, with most values remaining below 1.2%, further confirming the accuracy of the radar method compared to the traditional respiration belt sensor. Although there are slight variations in the percentages of errors between subjects, some showing higher errors under certain conditions, particularly in more dynamic environments such as driving and bumpy roads, these differences are minimal. The error percentages remain relatively stable under the different driving conditions, suggesting that the accuracy of the radar sensor is not significantly affected by the type of driving activity.

## 5. Discussion

The results from seat occupancy detection using the flexible transmitarray lens with 60 GHz FMCW radar provide valuable insights into the system’s ability to monitor seat occupancy and physiological signals, such as heart rate and breathing. The data demonstrated that the radar system reliably differentiated between an empty seat and an occupied one, with minimal noise interference, highlighting its reliability for this purpose. These findings align with previous studies that have shown the effectiveness of radar in capturing micro-movements like breathing and heart rate and in detecting the presence of occupants.

The broader implications of these results suggest that such noninvasive, radar-based monitoring systems with transmitarray lens could play a significant role in the future of smart vehicle technology. As automakers continue to develop safer and more autonomous vehicles, radar systems for occupancy detection, vital sign monitoring, and safety checks could become standard features. This capability not only improves passenger safety, but also provides continuous, real-time health monitoring.

Future research should focus on improving the robustness of the system in more dynamic driving conditions, such as bumpy roads or sudden changes in occupant behavior. Advanced noise reduction techniques with different algorithms could help isolate physiological signals from environmental noise. Furthermore, incorporating machine learning algorithms to distinguish between real and false detections, such as differentiating between a small object and a human occupant, could further enhance accuracy. Investigating the long-term durability and performance of the system in varying vehicle environments, including extreme weather, will be crucial to its widespread adoption.

## 6. Conclusions

Our study demonstrates the effectiveness of a seatbelt-integrated beam-focused metamaterial sensing system for the accurate detection of vital signs in an automotive environment. Integrating a flexible transmitarray lens with a 60 GHz FMCW radar sensor significantly improves the precision of physiological signal measurements. Validated against standard reference methods, the radar system closely tracks heart rate and respiration, underscoring its potential for improving in-car healthcare monitoring and is suitable for implementation on the seatbelt for seat occupancy detection. Future work will focus on further refining the robustness of the system and exploring broader applications.

## Figures and Tables

**Figure 1 sensors-24-07494-f001:**
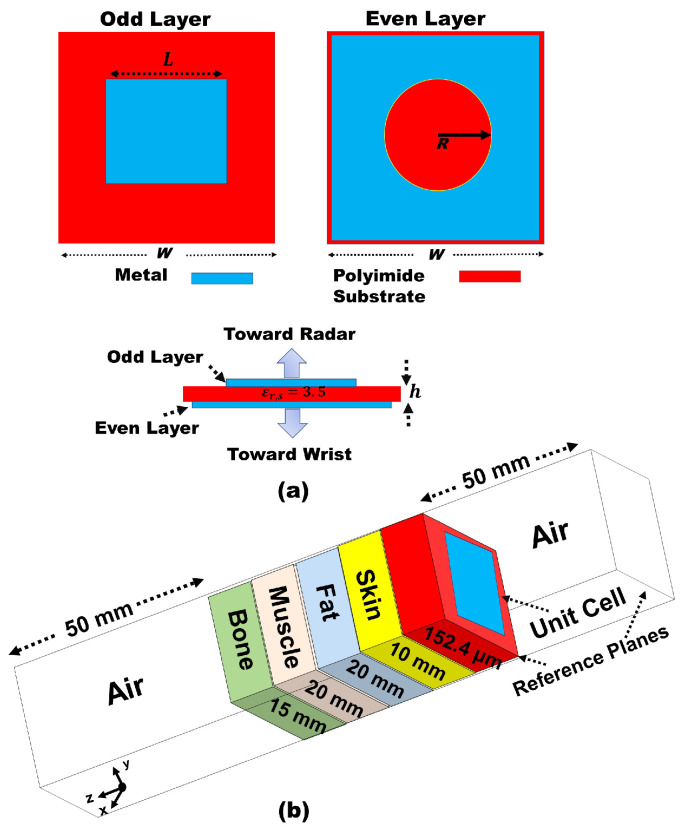
(**a**) Unit cell design of the transmitarray lens. (**b**) Unit cell setup in HFSS incorporating the human phantom model.

**Figure 2 sensors-24-07494-f002:**
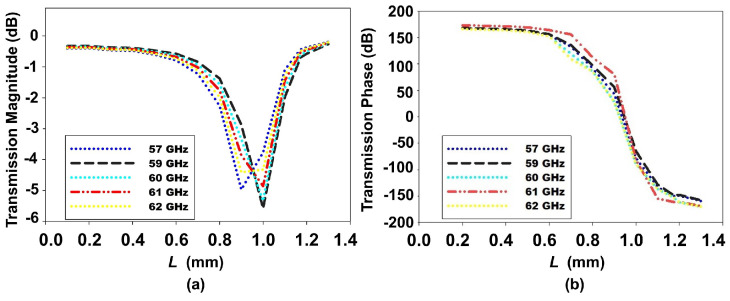
Transmission response at frequency 57–62 GHz. (**a**) Transmission magnitude for varying unit cell dimensions *L* = 0.15–1.4 mm, W = 1.6 mm, R = 0.6 mm. (**b**) Transmission phase.

**Figure 3 sensors-24-07494-f003:**
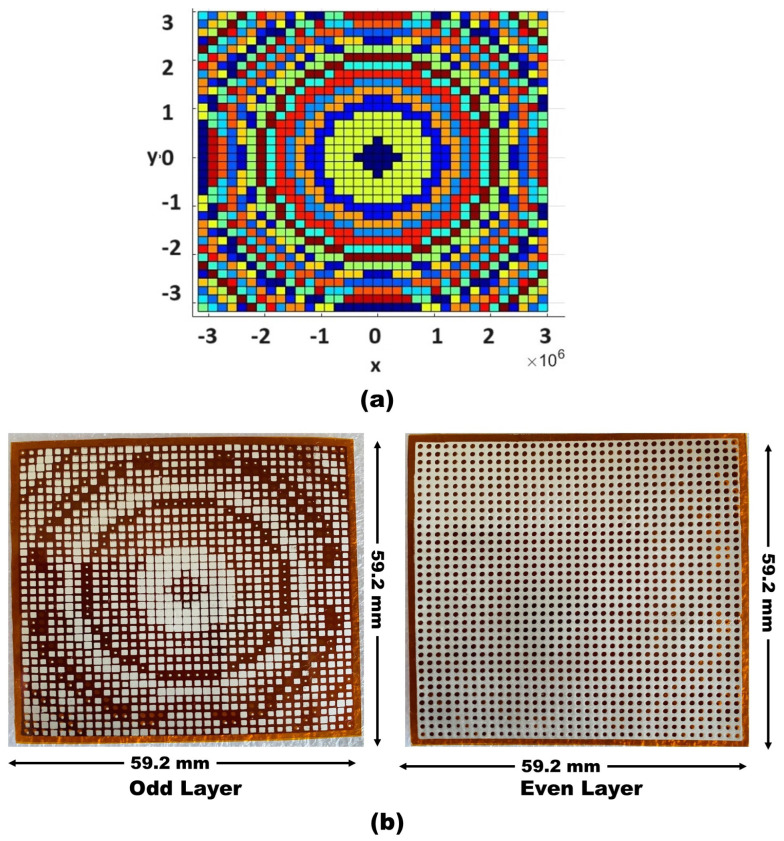
(**a**) Phase distribution optimization through MATLAB R2023b. (**b**) Fabricated metasurface odd–layer and even–layer.

**Figure 4 sensors-24-07494-f004:**
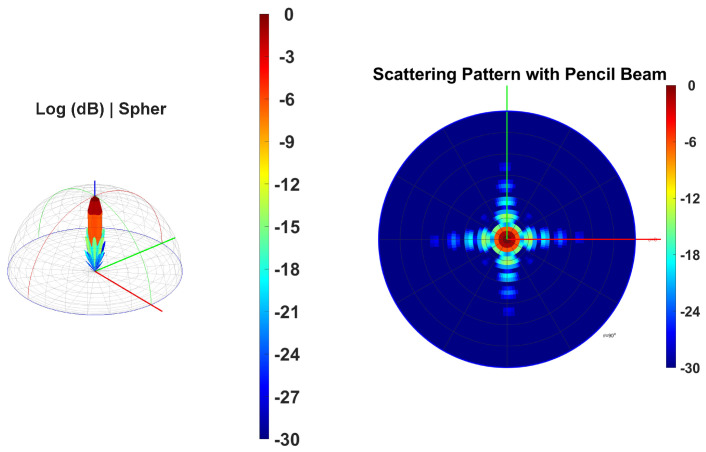
Metasurface beam direction in $0^{\circ }$ phase.

**Figure 5 sensors-24-07494-f005:**
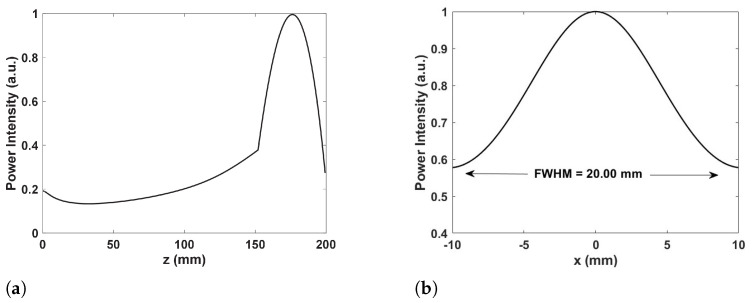
(**a**) The normalized power distributions along the z-axis in the center. The inset is the focused pattern on the x-y plane. (**b**) The simulated spot size FWHM of focused 60 GHz wave along the x-axis.

**Figure 6 sensors-24-07494-f006:**
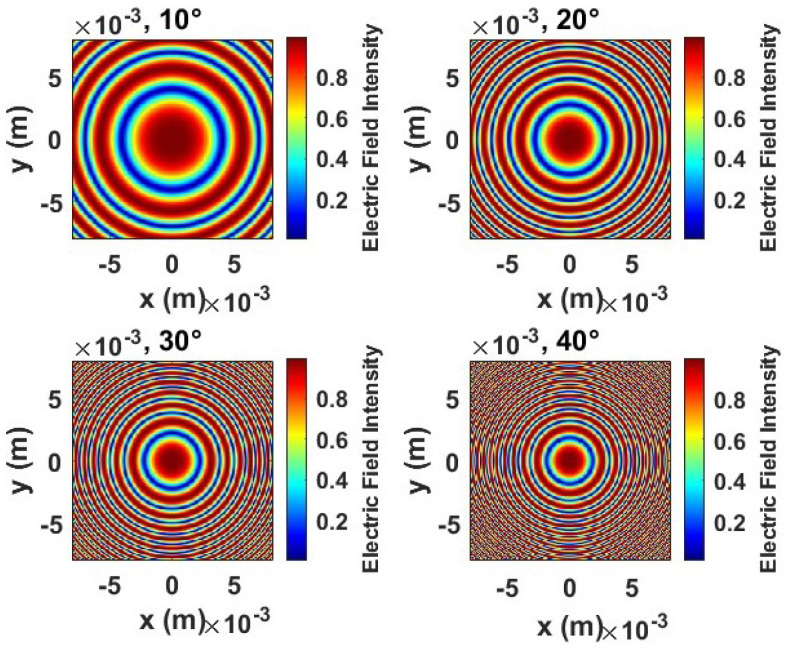
Different bending conditions through human tissue of the flexible transmitarray lens: $10^{\circ }$, $20^{\circ }$, $30^{\circ }$, and $40^{\circ }$.

**Figure 7 sensors-24-07494-f007:**
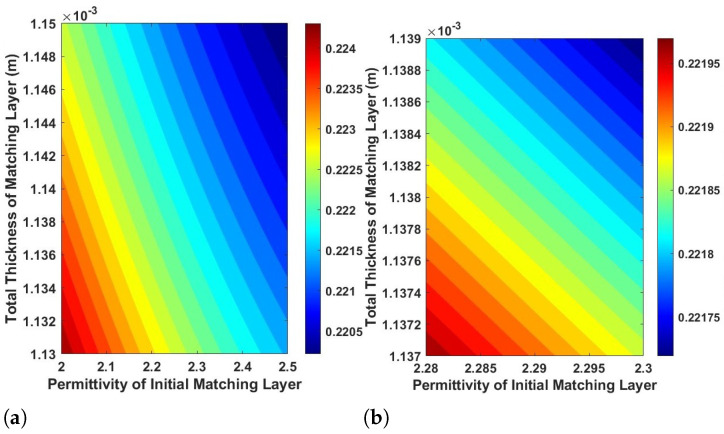
Comparison of matching layers showing reflection coefficients. (**a**) Comparison range reflection coefficient. (**b**) Optimal range reflection coefficient.

**Figure 8 sensors-24-07494-f008:**
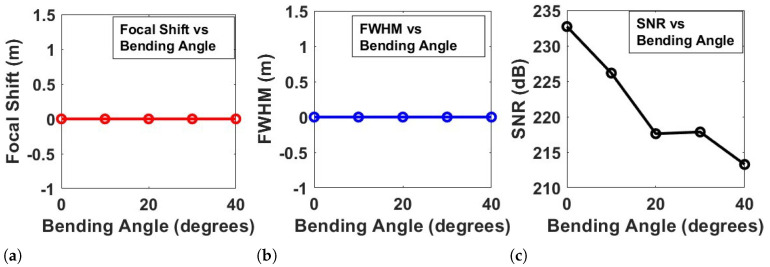
Simulation results for a flexible metasurface: (**a**) The focal shift vs. varying bending angles. (**b**) The FWHM vs. bending angle. (**c**) The SNR vs. bending angle.

**Figure 9 sensors-24-07494-f009:**
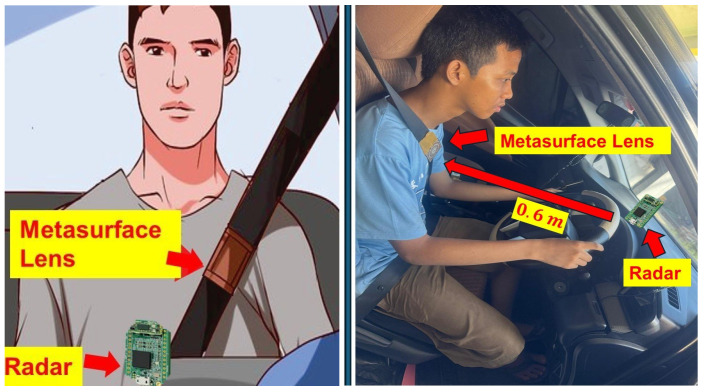
Experimental setup showing the integration of the metasurface lens with the seatbelt in-vehicle conditions.

**Figure 10 sensors-24-07494-f010:**
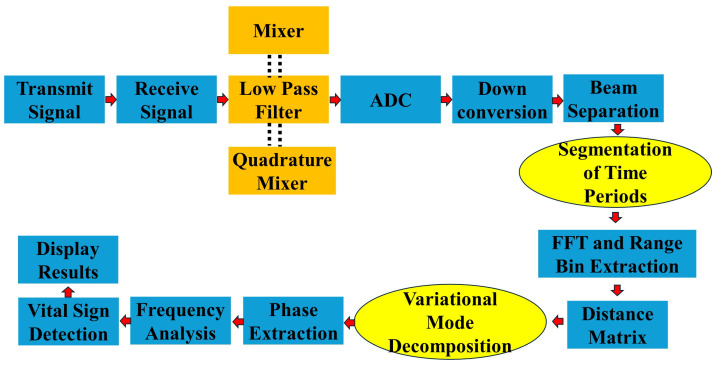
Signal Processing Block Diagram.

**Figure 11 sensors-24-07494-f011:**
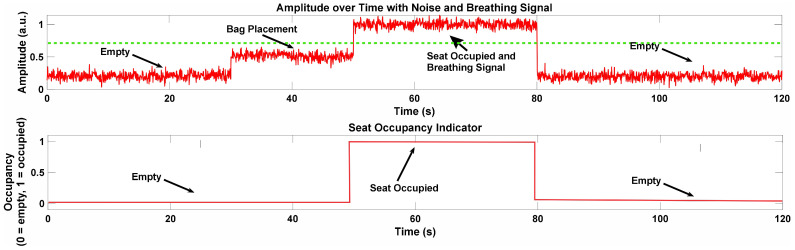
Seat occupancy detection using 60 GHz FMCW radar, showing range, amplitude (with noise and breathing signal), amplitude deviation, and seat occupancy indicator over time.

**Figure 12 sensors-24-07494-f012:**
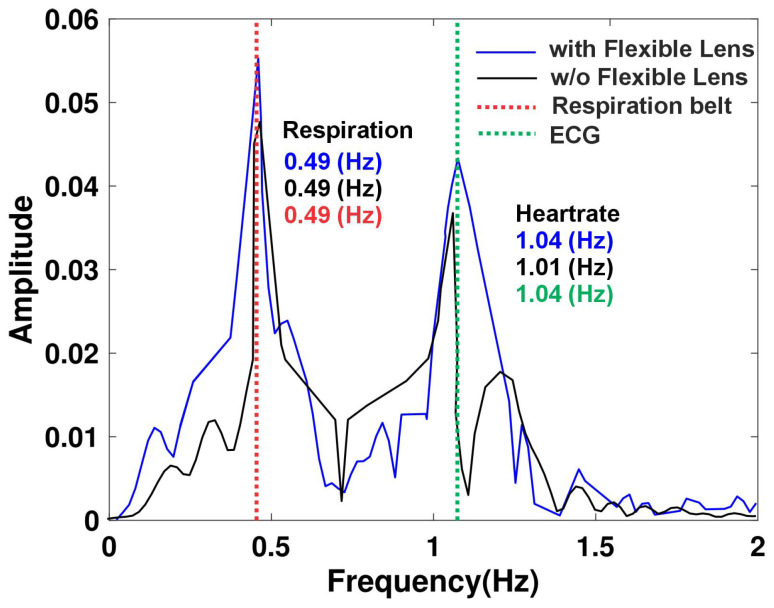
Heart rate and respiration comparison with or without the flexible lens.

**Figure 13 sensors-24-07494-f013:**
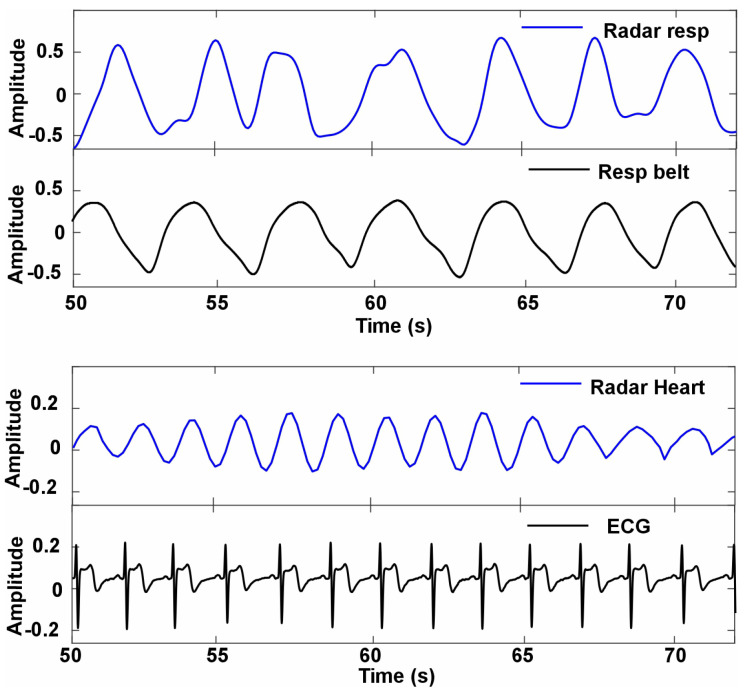
Heart rate and respiration comparison with flexible transmitarray lens.

**Figure 14 sensors-24-07494-f014:**
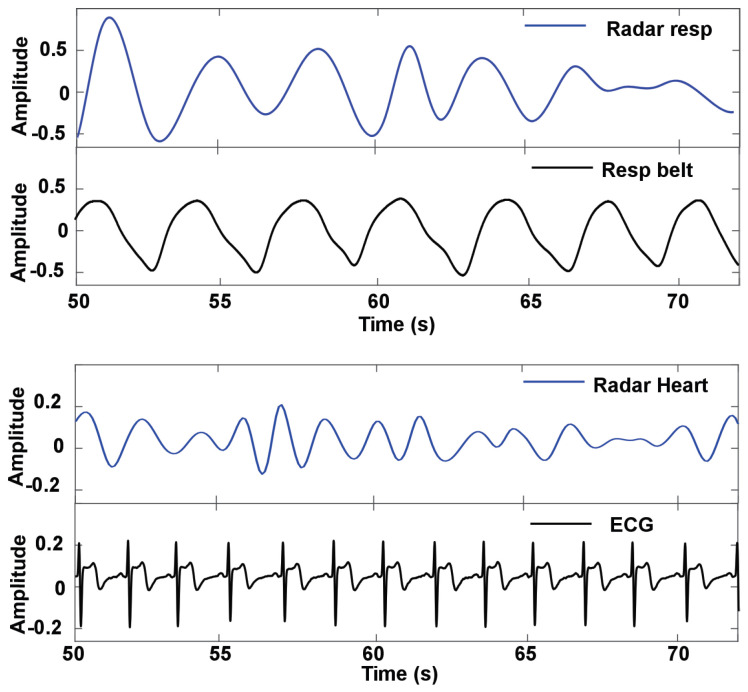
Heart rate and respiration comparison without flexible transmitarray lens.

**Figure 15 sensors-24-07494-f015:**
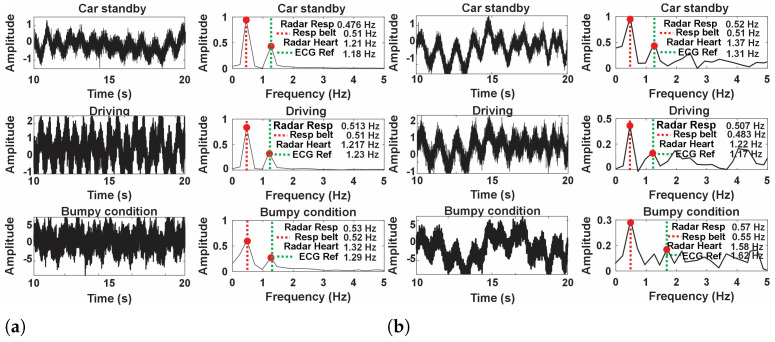
Comparison of heart rate and respiration detection under a different driving conditions such as standby, driving, and bumpy (**a**) with a flexible transmitarray lens and (**b**) without flexible transmitarray lens.

**Figure 16 sensors-24-07494-f016:**
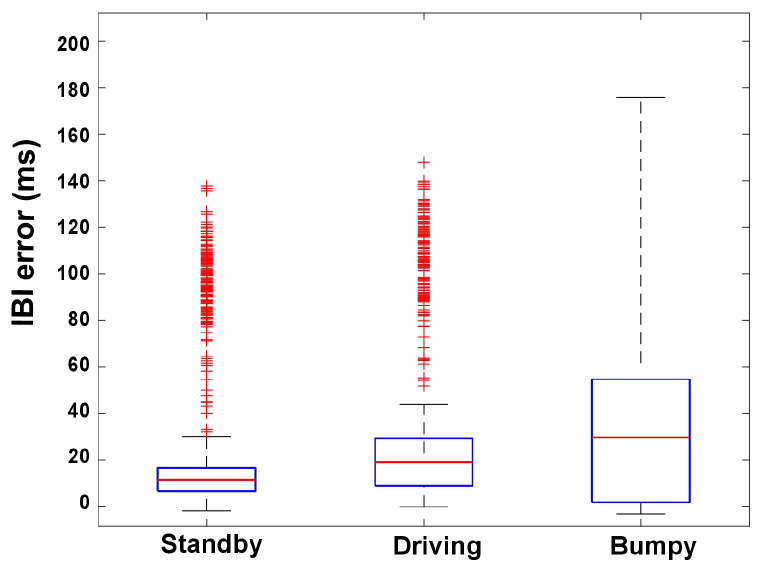
Comparison of inter−beat interval (IBI) errors during different driving conditions (standby, driving, and bumpy) using a flexible transmitarray lens.

**Figure 17 sensors-24-07494-f017:**
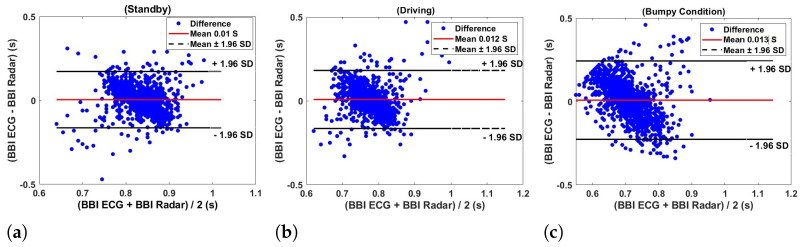
Bland–Altman plots showing the heart rate agreement between Beat-to-beat interval ECG and Radar measurements under different conditions from 5 subjects: (**a**) Car Standby, (**b**) Driving, and (**c**) Bumpy Condition using flexible transmitarray lens.

**Figure 18 sensors-24-07494-f018:**
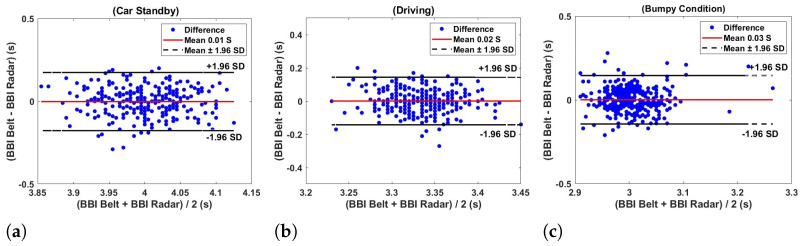
Bland–Altman plots showing the respiration rate agreement between breath-to-breath interval respiration belt and radar measurements from 5 subjects under different conditions such as (**a**) car standby, (**b**) driving, and (**c**) bumpy using a flexible transmitarray lens.

**Figure 19 sensors-24-07494-f019:**
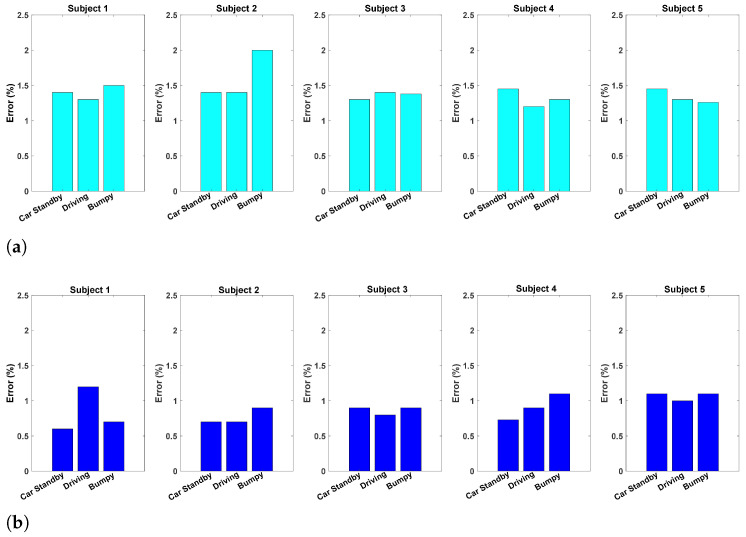
Error percentages of (**a**) heart rate and (**b**) respiration measurements under different conditions such as car standby, driving, and bumpy from 5 subjects using a flexible transmitarray lens.

**Table 1 sensors-24-07494-t001:** FMCW Radar parameters.

Parameter	Symbol	Value
Number of ADC samples	NTS	64
Number of Chirps	$N_{c}$	1
Chirp Time	$T_{c}$	32 μs
Transmit Antenna	$N_{TX}$	1
Receiver Antenna	$N_{RX}$	3
Total Bandwidth	*B*	5 GHz
Frame Time	$T_{f}$	75.476 ms
Azimuth Antenna Field of View	$\Theta_{\mathrm{FOV}}$	$70^{\circ }$

**Table 2 sensors-24-07494-t002:** Comparison of RMS acceleration, heart rate coefficient of variation (CV), and respiration coefficient of variation across conditions with and without the flexible transmitarray lens.

Condition	With Lens	Without Lens
	RMS Acc./CV HR/CV Resp.	RMS Acc./CV HR/CV Resp.
Standby	0.030/0.011/12.20	0.037/0.021/14.20
Driving	0.200/0.021/13.00	0.325/0.031/14.26
Bumpy	0.310/0.045/14.30	0.470/0.075/15.10

## Data Availability

Not applicable.

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
