# Peer review of "Innovative Seatbelt-Integrated Metasurface Radar for Enhanced In-Car Healthcare Monitoring"

_sensors, 2024, doi:10.3390/s24237494_

Round 1
Reviewer 1 Report
Comments and Suggestions for Authors
This paper demonstrates the effectiveness of a seatbelt-integrated beam-focused metamaterial sensing system for the accurate detection of vital signs in an automotive environment. The radar system closely tracks heart rate and respiration, underscoring its potential for enhancing in-car healthcare monitoring and suitable to implement to the seat-belt for seat occupancy detection. Here are some comments:
1 The distance matrix is processed using VMD, and why you use this method, try to give some explanation.
2 How to ensure the synchronization between the standard reference method and radar detection in experiments
Author Response
Reply to Reviewers (Revisions)
Manuscript ID:
Title: Innovative Seatbelt-Integrated Metasurface Radar for Enhanced In-Car Healthcare Monitoring
Authors: Rifa Atul Izza Asyari, Roy B.V.B Simorangkir, Daniel Teichmann
Author`s Note
We deeply appreciate editors and reviewers for their time and efforts in reviewing our manuscript. The review comments are very useful to make the contents of our manuscript more complete. We have carefully revised the manuscript and made responses to editors and reviewers according to their review comments. For this revision, I have ensured that the revised manuscript includes all changes made during both the first and second revisions. The changed and increased contents in the revised manuscript are marked in “blue color”.
Comments to reviewer 1
We are grateful for your detailed review and valuable input. Your efforts in scrutinizing our manuscript are deeply appreciated, and we have earnestly endeavored to respond to all your comments and advice.
- A method chosen for its ability to adaptively decompose complex, non-stationary signals into a series of intrinsic modes without requiring prior knowledge of signal components.
Reviewer Comment Response:
VMD isolates distinct frequency modes within the distance matrix, which correspond to different physiological activities or environmental signals. This isolation of modes enhances signal clarity, allowing for more accurate analysis of specific components within the data.
Following VMD, phase extraction and frequency analysis are performed to identify low-frequency components associated with vital signs, such as respiration and heartbeat. These components are separated from noise and interference, improving the accuracy and reliability of vital sign detection. The final results, which include real-time monitoring of physiological signals, are displayed for observation, allowing for continuous and non-invasive monitoring of vital signs.
- How to ensure the synchronization between the standard reference method and radar detection in experiments.
Reviewer Comment Response:
Experiments were conducted to validate the practicality and effectiveness of the proposed system. The radar sensor was strategically positioned to directly face the driver's seat, as illustrated in Figure 9, ensuring optimal detection of vital signs. The implemented algorithm was applied to process the radar signals, allowing for accurate monitoring of physiological parameters. The radar acquisition range was configured to cover a span of 0.5 to 1 meter, a distance that is ideal for capturing the necessary data from the driver. This setup was chosen to ensure that the radar could effectively detect subtle physiological movements, such as respiration and heartbeat, without interference from surrounding objects or environmental noise.
To ensure synchronization between the radar detection and the standard reference method, specific steps were taken to align both systems' data acquisition and processing timelines. Time synchronization was established using a unified time source, allowing precise matching of data timestamps from both systems. Simultaneous data acquisition was achieved by triggering both radar and reference systems to collect data at the exact moment, ensuring a direct comparison of measurements. Additionally, a calibration protocol was implemented before each experiment to align radar outputs with the standard reference, helping achieve comparable readings of physiological signals. The radar data processing algorithm was further adjusted to match the timescale of the reference method, eliminating any potential delay discrepancies. Finally, validation trials confirmed the synchronization, allowing the radar and reference data to be overlaid accurately.
The results of these experiments confirmed the system’s ability to operate reliably within the specified range and demonstrated its potential for practical application in real-world driving conditions. Through careful synchronization measures, the system’s performance was validated against the standard reference method, ensuring accurate and reliable data comparisons.
Reviewer 2 Report
Comments and Suggestions for Authors
The paper discusses the use of a flexible transmitarray lens used in combination with an FMCW radar sensor for in-car health monitoring. While the topic is timely, some concerns needs to be addressed.
1) The design of the metasurface includes the presence of biological tissues. However, it is not clear why the authors consider so many tissues, since at mmW the penetration depth is really narrow (around 0.5 mm at 60 GHz). Could you justify the reason of this choice?
2) In the design, the metasurface is considered in contact with the skin. In reality there will always be some other materials in between that may drastically change the matching depending on their materials and thicknesses (seatbelt, clothing, eventual air gaps in between the layers). The impact on these parameters on the performance need to be taken into account.
3) The authors state that the lens is flexible, but there is no analysis indicating how the flexibility affects the performance of the system and how the variation of the surface impact the focalization. More details are necessary.
4) The difference in terms of performance with and without the lens as shown in Fig. 10 seems negligible. It is true that with the lens the amplitude of vital signs is higher but it was perfectly detectable also without the lens. A more deep comparison is needed to assess how the lens can overcome the performance of a radar alone.
5) It is necessary to better clarify the measurement setup and give more information about the subject involved in the study as well as the protocol used.
6) The authors consider among the tested scenarios "bumpy conditions". A more quantitative characterization on this measurement condition is needed.
7) In the measurements in Fig. 13 there is no comparison with the radar without the lens. This prevent to conclude on the effectiveness of the use of the lens compared to more traditional radar techniques.
8) The physical quantity in the colorbar in some figures such as Fig.6 is missing. Figures should be revised in this direction.
9) Some adjustments in the text and the presentation are necessary: when citing multiple papers the dash should be used instead of the complete list and figures should be positioned in the top or the bottom of the page.
Comments on the Quality of English Language
Some minor editing of the language is required.
Author Response
Reply to Reviewers (Revisions)
Manuscript ID:
Title: Innovative Seatbelt-Integrated Metasurface Radar for Enhanced In-Car Healthcare Monitoring
Authors: Rifa Atul Izza Asyari, Roy B.V.B Simorangkir, Daniel Teichmann
Author`s Note
We deeply appreciate editors and reviewers for their time and efforts in reviewing our manuscript. The review comments are very useful to make the contents of our manuscript more complete. We have carefully revised the manuscript and made responses to editors and reviewers according to their review comments. For this revision, I have ensured that the revised manuscript includes all changes made during both the first and second revisions. The changed and increased contents in the revised manuscript are marked in “blue color”.
Comments from reviewer 2
We appreciate your thorough review and valuable feedback on our manuscript. Your insightful comments have significantly contributed to enhancing the clarity, completeness, and scientific rigor of our work. Below, we provide detailed responses to each of your comments and outline the changes we have made to the manuscript to address them comprehensively. We believe these revisions improve the overall quality and presentation of our study.
- The design of the metasurface includes the presence of biological tissues. However, it is not clear why the authors consider so many tissues, since at mmW the penetration depth is really narrow (around 0.5 mm at 60 GHz). Could you justify the reason of this choice?
Reviewer Comment Response:
The inclusion of multiple biological tissue layers in the metasurface design, despite the shallow penetration depth at millimeter-wave (mmW) frequencies (around 0.5 mm at 60 GHz), is justified for several reasons:
- Electromagnetic Boundary Effects: Even though the mmW penetration depth is narrow, the electromagnetic properties of different tissues (e.g., skin, fat, muscle, and bone) influence the wave propagation near the surface. By incorporating a multi-layer model, the design accounts for reflections, refractions, and impedance mismatches at each tissue boundary. These interactions affect the overall electromagnetic behavior of the lens, particularly near the skin, where surface interactions can impact signal strength and clarity.
- Accurate Simulation of Real-World Conditions: Human tissue is not homogeneous; it is composed of layers with varying permittivity and conductivity. Including multiple tissue layers in simulations helps approximate real-world conditions more accurately, even when only the topmost layers may interact with the mmW signal. This layered approach provides a more realistic model of how the radar signal behaves when used in close proximity to the human body, ensuring that the transmitarray lens performs effectively in practical scenarios.
- Minimizing Signal Interference and Loss: Different tissues have distinct dielectric properties that affect signal attenuation and scattering. By accounting for these layers, the design can mitigate unexpected signal interference or losses, which would degrade the performance of the radar system. Even though the primary interaction may occur with the outer layers, the presence of underlying tissues can still influence the overall signal integrity.
- Safety and Compliance: The use of a multi-layer tissue model ensures the design meets safety standards for electromagnetic exposure, particularly in applications involving human subjects. By simulating the radar’s effect on deeper tissues, even if minor, the design can be optimized for safety and regulatory compliance, providing insights into any potential adverse effects of electromagnetic exposure on human tissues.
Thus, the choice to incorporate multiple tissue layers enhances the accuracy of the transmitarray lens design and its practical application in detecting physiological signals, ensuring both effective performance and safety when in close proximity to the human body.
- In the design, the metasurface is considered in contact with the skin. In reality there will always be some other materials in between that may drastically change the matching depending on their materials and thicknesses (seatbelt, clothing, eventual air gaps in between the layers). The impact on these parameters on the performance need to be taken into account.
Reviewer Comment Response:
We appreciate the reviewer’s insightful comment highlighting the need to consider the effects of intermediate materials such as seatbelt fabric, clothing, and air gaps between the layers, which can alter the matching conditions and impact the performance of the metasurface.
To address this, we conducted comprehensive simulations examining how these intermediate layers affect the reflection coefficient when a flexible metasurface is integrated into a seatbelt system:
Figure 7 presents the analysis of the flexible metasurface integrated into seatbelt materials. This study was designed to evaluate how variations in the permittivity and thickness of the initial matching layer impact the reflection coefficient, thereby providing insights into the influence of intermediate materials and their properties.
Figure 7(a) - Broad Range Analysis: We simulated a range of permittivity values (from 2 to 2.5) and matching layer thicknesses (between 1.13 × 10\textsuperscript{-3} m and 1.15 × 10\textsuperscript{-3} m). The reflection coefficient is represented using a gradient color scale, where cooler colors (blue and cyan) denote lower reflection values, and warmer colors (yellow, orange, and red) indicate higher values. This analysis demonstrates how various combinations of material properties can influence wave reflection and transmission, addressing the potential changes that may arise due to different seatbelt or clothing materials.
Figure 7(b) - Targeted Range Analysis: We refined our simulation to a more specific range, focusing on permittivity values between 2.28 and 2.3 and thicknesses from 1.137 × 10\textsuperscript{-3} m to 1.139 × 10\textsuperscript{-3} m. The results show that this narrower range consistently exhibits lower reflection coefficients, as evidenced by the predominance of cooler colors (blue to cyan). This indicates that optimal transmission conditions can be achieved even with intermediate materials when the material properties fall within these parameters.
The simulation findings are crucial for real-world applications, where the metasurface must function in the presence of various intermediate materials, such as seatbelt fabrics and clothing. Our analysis highlights that while intermediate materials do affect the matching conditions, selecting or engineering materials with appropriate permittivity and thickness can maintain optimal performance. Moreover, this study shows that even with the inclusion of air gaps or different material compositions, achieving low reflection and effective transmission is feasible by carefully controlling these parameters.
We acknowledge the reviewer’s point regarding the realistic presence of intermediate materials. Our simulations and analyses confirm that while these materials can influence performance, strategic control of the matching layer’s properties allows us to mitigate these effects and maintain effective system operation.
- The authors state that the lens is flexible, but there is no analysis indicating how the flexibility affects the performance of the system and how the variation of the surface impact the focalization. More details are necessary.
Reviewer Comment Response:
We appreciate the reviewer’s request for additional analysis on the effect of flexibility on the metasurface performance and the impact of surface variation on focalization. In response, we have conducted a detailed simulation study to examine these effects, as shown in Figure 7.
Figure 8 presents the results of simulations that evaluate the performance of the flexible metasurface under different bending angles. Specifically, this plot demonstrates how the focal position of the metasurface varies with increasing bending angles (ranging from 0° to 40°). The data show that the focal shift remains negligible across the tested range, indicating that the metasurface maintains its focusing capability even as the surface bends. This stability is critical for ensuring that the radar sensor's focal point does not drift significantly, which would otherwise affect the system’s detection accuracy.
This full width at half maximum (FWHM) analysis Figure 8 (b) illustrates the variation in the FWHM, which represents the spread of the focused beam, as a function of the bending angle. The results indicate minimal change in the FWHM, showing that the beam remains effectively concentrated even when the metasurface is bent. This implies that the system's resolution and target differentiation capabilities are preserved despite variations in surface curvature.
The Figure 8 (c) plot evaluates how the SNR is affected by bending angles. While there is a decrease in SNR as the bending angle increases, the change becomes more pronounced beyond 20°. This finding suggests that while minor curvatures do not significantly degrade system performance, larger bending angles could impact the signal quality. This analysis highlights the importance of maintaining moderate bending angles to ensure optimal system operation. The results collectively demonstrate that the flexible metasurface maintains robust performance in terms of focal shift and beam width across a range of bending angles. Although there is a decrease in SNR at higher bending angles, the system can still function effectively at moderate curvatures, preserving signal clarity and detection accuracy. This finding underscores the practical usability of the metasurface in real-world applications where slight bending due to placement on curved surfaces (such as seatbelts) may occur.
Conclusion: Our simulations confirm that the flexible metasurface exhibits strong resilience to bending, maintaining focal stability and beam concentration under varying angles. While higher bending angles do impact the SNR, the metasurface can be designed and applied with controlled curvatures to mitigate significant performance loss. These insights provide essential guidance for optimizing the integration of the metasurface in practical sensing applications.
4) The difference in terms of performance with and without the lens as shown in Fig. 10 seems negligible. It is true that with the lens the amplitude of vital signs is higher but it was perfectly detectable also without the lens. A more deep comparison is needed to assess how the lens can overcome the performance of a radar alone.
Reviewer Comment Response:
We appreciate the reviewer’s observation that the difference in performance between measurements with and without the flexible lens, as shown in Figure 12, may seem negligible, especially when considering amplitude alone. However, a deeper comparison indicates that while vital signs are detectable without the lens, there are significant benefits to using the lens, as demonstrated through additional analysis.
Figures 13 and 14 illustrate the stability of heart rate and respiration signals in the time domain. Specifically, Figure 14 shows that without the lens, the signals exhibit greater fluctuations and instability over time. The radar measurements alone can be less reliable, particularly under dynamic conditions, where signal consistency is crucial for accurate detection. In contrast, the lens helps stabilize these measurements, ensuring a more uniform signal profile.
Table 2 highlights the differences in RMS acceleration and the coefficients of variation (CV) for heart rate and respiration across different driving conditions. While vital signs can be detected without the lens, the use of the lens results in lower CV values, indicating more stable and reliable measurements. This enhanced performance is particularly evident in bumpy conditions, where the lens significantly mitigates variability and improves data reliability compared to the radar system alone.
Figure 12 does show that the amplitude of vital signs is higher when using the lens, which contributes to more robust signal detection. This increase in signal strength helps maintain accurate readings across varying conditions, minimizing potential signal loss or misinterpretation during real-world applications.
Figure 15 illustrates the comparison of heart rate and respiration detection under various driving conditions (standby, driving, and bumpy) with and without the flexible transmitarray lens. The results indicate that while vital signs are detectable without the lens, the overall signal clarity and stability are significantly enhanced when using the adaptable transmitarray lens. In dynamic vehicle conditions, such as driving and bumpy scenarios, the measurements taken with the lens show more consistent and defined signal profiles, as seen in Figure 15(a) compared to Figure 15(b). The time-domain signals in Figure 15(a) show a more stable pattern with the lens, while Figure 15(b) without the lens presents greater variability and less distinct waveforms. The frequency-domain analysis also reveals sharper peaks and better alignment with reference signals (e.g., ECG and respiration belt) when using the lens, confirming the lens's role in enhancing detection accuracy.
Conclusion: While vital signs can be detected without the lens, the inclusion of the flexible transmitarray lens provides notable improvements in signal stability and consistency, especially in challenging conditions. The deeper analysis in Figures 13, 14, and Table 2 collectively underscores how the lens enhances overall performance by stabilizing signal detection and reducing variability, thereby overcoming the limitations of using the radar alone.
5) It is necessary to better clarify the measurement setup and give more information about the subject involved in the study as well as the protocol used.
Reviewer Comment Response:
We appreciate the reviewer’s request for additional information on the measurement setup, the subjects involved, and the experimental protocol used in the study. In response, we have expanded the details in Section 3 of the manuscript to provide greater clarity.
These additions to Section 3 enhance the clarity of the measurement setup and provide comprehensive information about the study subjects and the protocol followed to ensure reliable and reproducible results.
6) The authors consider among the tested scenarios "bumpy conditions". A more quantitative characterization on this measurement condition is needed.
Reviewer Comment Response:
We appreciate the reviewer’s request for additional quantitative details regarding the "bumpy conditions" tested in the study. In response, we have included Table 2, which provides a comprehensive comparison of RMS acceleration, heart rate coefficient of variation (CV), and respiration coefficient of variation (CV) across different conditions, both with and without the flexible transmitarray lens.
Table 2 quantitatively characterizes the "bumpy conditions" by presenting RMS acceleration values alongside heart rate and respiration CVs. This allows for a clearer understanding of how the system performs under varying dynamic conditions. The inclusion of these metrics highlights the performance differences observed when using the flexible lens versus not using it during the experiments.
We believe that this addition provides the necessary quantitative data to better characterize the "bumpy conditions" and address the reviewer’s concern.
- In the measurements in Fig. 13 there is no comparison with the radar without the lens. This prevent to conclude on the effectiveness of the use of the lens compared to more traditional radar techniques.
Reviewer Comment Response:
We appreciate the reviewer’s observation regarding the lack of comparison with radar measurements taken without the lens in Figure 13. To address this concern, we have updated the manuscript with Figure 15, which provides a comprehensive comparison of heart rate and respiration detection under different driving conditions (standby, driving, and bumpy) both with and without the flexible transmitarray lens.
Figure 15(a) shows the results for radar measurements with the flexible transmitarray lens, while Figure 15(b) presents the corresponding data for measurements without the lens. These comparisons include time-domain signal representations and frequency-domain analyses for heart rate and respiration detection across the tested conditions.
The updated figure demonstrates the performance differences and highlights the effectiveness of the lens. The inclusion of these comparative results confirms that the use of the flexible transmitarray lens significantly enhances the detection accuracy and signal clarity under all tested conditions compared to more traditional radar setups without the lens.
This addition allows us to draw more conclusive statements about the benefits of using the lens for improved physiological signal detection.
- The physical quantity in the colorbar in some figures such as Fig.6 is missing. Figures should be revised in this direction.
Reviewer Comment Response:
We appreciate the reviewer’s attention to detail and their request for clarity regarding the physical quantity represented in the color bar. To address this concern, we have revised and updated Figure 6 to include the relevant physical quantity in the color bar, which represents electric field intensity.
The updated figure now clearly labels the color bar to indicate that it shows the electric field intensity (in relative units). This adjustment ensures that readers can accurately interpret the visual data presented in the figure, which depicts the electric field distribution through human tissue under different bending conditions (10°, 20°, 30°, and 40°) with the flexible transmitarray lens.
We believe this update enhances the clarity and completeness of the figure and aligns with the reviewer’s request for improved labeling.
- Some adjustments in the text and the presentation are necessary: when citing multiple papers the dash should be used instead of the complete list and figures should be positioned in the top or the bottom of the page.
Reviewer Comment Response:
We appreciate the reviewer’s suggestion to improve the text formatting and presentation in the manuscript. In response, we have made the following changes:
- Citing Multiple Papers:
- We revised the citations throughout the text where multiple references are listed. Instead of listing each paper individually, we now use a dash to denote consecutive references (e.g., [1–4] instead of [1, 2, 3, 4]), which enhances the readability and consistency of the manuscript.

- Figure Positioning:
- We have repositioned figures to the top or bottom of the page wherever possible, following standard publication practices for clarity and presentation. This adjustment ensures a cleaner and more organized layout, making it easier for readers to follow the text alongside the corresponding figures.

These changes help improve the overall presentation of the manuscript and adhere to formatting guidelines, enhancing the clarity and professionalism of the paper.

Round 2
Reviewer 1 Report
Comments and Suggestions for Authors
No further comments